# Cancer Diagnosis through Contour Visualization of Gene Expression Leveraging Deep Learning Techniques

**DOI:** 10.3390/diagnostics13223452

**Published:** 2023-11-15

**Authors:** Vinoth Kumar Venkatesan, Karthick Raghunath Kuppusamy Murugesan, Kaladevi Amarakundhi Chandrasekaran, Mahesh Thyluru Ramakrishna, Surbhi Bhatia Khan, Ahlam Almusharraf, Abdullah Albuali

**Affiliations:** 1School of Computer Science Engineering and Information Systems (SCORE), Vellore Institute of Technology, Vellore 632014, India; drvinothkumar03@gmail.com; 2Department of Computer Science and Engineering, Faculty of Engineering and Technology, JAIN (Deemed-to-be University), Bangalore 562112, India; karthick.km@jainuniversity.ac.in (K.R.K.M.); trmahesh.1978@gmail.com (M.T.R.); 3Department of Computer Science and Engineering, Sona College of Technology, Salem 636005, India; kaladeviac@sonatech.ac.in; 4Department of Data Science, School of Science Engineering and Environment, University of Salford, Manchester M5 4WT, UK; 5Department of Engineering and Environment, University of Religions and Denominations, Qom 37491-13357, Iran; 6Department of Electrical and Computer Engineering, Lebanese American University, Byblos P.O. Box 13-5053, Lebanon; 7Department of Business Administration, College of Business and Administration, Princess Nourah bint Abdulrahman University, Riyadh 11671, Saudi Arabia; aialmusharraf@pnu.edu.sa; 8Department of Computer Science, School of Computer Science and Information Technology, King Faisal University, Hofuf 11671, Saudi Arabia; aalbuali@kfu.edu.sa

**Keywords:** accuracy, classification, detection, diagnosis, contour, visualization, computation, cancer, loss, precision, recall

## Abstract

Prompt diagnostics and appropriate cancer therapy necessitate the use of gene expression databases. The integration of analytical methods can enhance detection precision by capturing intricate patterns and subtle connections in the data. This study proposes a diagnostic-integrated approach combining Empirical Bayes Harmonization (EBS), Jensen–Shannon Divergence (JSD), deep learning, and contour mathematics for cancer detection using gene expression data. EBS preprocesses the gene expression data, while JSD measures the distributional differences between cancerous and non-cancerous samples, providing invaluable insights into gene expression patterns. Deep learning (DL) models are employed for automatic deep feature extraction and to discern complex patterns from the data. Contour mathematics is applied to visualize decision boundaries and regions in the high-dimensional feature space. JSD imparts significant information to the deep learning model, directing it to concentrate on pertinent features associated with cancerous samples. Contour visualization elucidates the model’s decision-making process, bolstering interpretability. The amalgamation of JSD, deep learning, and contour mathematics in gene expression dataset analysis diagnostics presents a promising pathway for precise cancer detection. This method taps into the prowess of deep learning for feature extraction while employing JSD to pinpoint distributional differences and contour mathematics for visual elucidation. The outcomes underscore its potential as a formidable instrument for cancer detection, furnishing crucial insights for timely diagnostics and tailor-made treatment strategies.

## 1. Introduction

Cancer is another notable driver of fatalities everywhere, behind cardiovascular disease. The World Health Organization (WHO) reported that more than nine million individuals died from cancer in 2018, with the number of new cases projected to rise to twenty-seven million annually by 2040 (World Health Organization (WHO), 2019). An accurate and prompt cancer diagnosis is thus essential for improving rates of recovery and survival via patient outcomes.

Due to the rising abundance of gene expression data from populations worldwide, there has never been a better opportunity to uncover the molecular insights of cancer. However, novel approaches are required for efficient analysis and interpretation due to these data’s extensive dimensionality and multifaceted nature [1].

Cancer continues to be a global health challenge, causing substantial morbidity and mortality worldwide. Early and accurate cancer detection is critical for successful treatment outcomes and predictive analytics [2], making integrating multiple analytical techniques essential for enhancing diagnostic accuracy [3]. In this context, gene expression datasets have emerged as a valuable resource, providing insights into the molecular mechanisms underlying cancer development and progression.

The need of the hour is to explore innovative methodologies that harness the potential of gene expression data for cancer detection. Grasping the intricate patterns and associations in these high-dimensional data can greatly enhance our comprehension of cancer biology and aid in tailoring individualized treatment approaches [4]. Integrating multiple analytical techniques offers a unique opportunity to extract valuable information from gene expression datasets, paving the way for more effective and timely cancer diagnosis and intervention.

The integration of EBS [5], JSD, deep learning, and contour mathematics in the analysis of gene expression data presents a cutting-edge and innovative approach in the process of cancer detection. Combining these powerful analytical techniques, this research holds significant potential to revolutionize cancer diagnosis and treatment strategies worldwide. JSD [6] enables the measurement of distributional differences between cancerous and non-cancerous samples, providing valuable insights into gene expression patterns associated with cancer. Deep learning models bring automatic feature extraction and pattern recognition capabilities, enabling the discovery of complex molecular relationships crucial for accurate cancer detection [7]. Additionally, contour mathematics offers a unique visualization tool, aiding in the interpretation of the model’s decision boundaries and regions in the high-dimensional feature space [8]. Ultimately, this integrated approach aims to contribute to the global fight against cancer by improving early diagnosis and enabling personalized treatment approaches, thus alleviating the burden of cancer on a global scale.

This research presents a novel and integrated approach to cancer detection, combining EBS, JSD, deep learning, and contour mathematics in the analysis of gene expression data. The objectives of this study are three-fold:

To leverage the power of EBS and JSD as an information-theoretic measure to quantify distributional differences between cancerous and non-cancerous samples based on preprocessed data. By integrating JSD into the analysis, this research aims to gain deeper insights into gene expression patterns, enabling the identification of critical genomic signatures associated with cancer.To harness the capabilities of deep learning models for automatic feature extraction and pattern recognition from gene expression data. By employing deep learning, this research seeks to uncover complex molecular relationships and identify crucial features that contribute to accurate cancer detection.To utilize contour mathematics for visual interpretation of the deep learning model’s decision boundaries and regions in the high-dimensional feature space. This novel visualization approach enhances the interpretability of the model, facilitating a deeper understanding of the complex interactions between genes and their relevance in cancer detection.

The overall outline of the research strategy is sequenced as follows. Section 2 reviews the most recent research progress carried out on gene expression cancer datasets, Section 3 delineates the characteristics of the incorporated dataset, Section 4 elaborates the methodology with the essential computational process, Section 5 assesses and discusses the observed empirical outcome of the proposed strategy, and lastly, Section 6 presents a brief conclusion and notes on the attained objective with possible future enhancements.

## 2. Related Work

This study [9] compare three deep learning methods, namely MLP, 1DCNN, and 2DCNN, in terms of their effectiveness in processing different cancer gene expression datasets. Using both balanced and unbalanced datasets, the study utilizes feature selection strategies such as ANOVA and Information Gain. Notably, 1DCNN outperforms others in terms of F1-score and accuracy, despite MLP showing superior performance in terms of False Positive Rate. The results highlight the promising potential of deep learning techniques in gene expression datasets.

This work [10] presented DEGnext, a convolutional neural network model designed for predicting upregulated (UR) and downregulated (DR) genes using gene expression data from The Cancer Genome Atlas. Utilizing transfer learning, the model leveraged insights from training feature maps on new, untrained cancer datasets. Compared to five traditional machine learning techniques, DEGnext demonstrated strong performance (with ROC scores ranging between 88% and 99%), establishing its reliability and efficiency in transferring learned features for new data classification and verifying the connection of major cancer-related Gene Ontology terms and pathways to the differentially expressed genes (DEGs) predicted by DEGnext.

The authors in this [11], explored eight supervised machine learning methods for cancer classification using the TCGA PancancerHiSeq dataset, which includes various types of cancers. The study involved preprocessing steps, including feature selection, oversampling, and normalization. The study showcases the utility of machine learning in improving the diagnostic precision of diverse cancers through effective feature selection and balancing of the dataset.

This study [12], proposed an optimized DL approach for classifying different types of cancers using tumor RNA sequence data. The study leveraged a novel combination of binary particle swarm optimization with a decision tree (BPSO-DT) and convolutional neural network (CNN), converting high-dimensional RNA-seq data into 2D images, which were then enhanced using an augmentation strategy. The study demonstrated impressive performance, achieving a classification accuracy of 96.90% across five cancer types, showing improved memory efficiency and simplicity in implementation.

This work [13], leveraged a part of the Pan-Cancer dataset to pre-train convolutional neural networks (CNNs) for predicting survival rates in lung cancer. The researchers tackled the lack of structure in gene expression data by reformatting RNA-seq samples into gene expression images, which enabled the extraction of high-level features through CNNs. They also investigated if integrating data from various tumor types could enhance the predictability of lung cancer progression compared to other machine learning methods. Ref. [14] introduced multiple convolutional neural network (CNN) models for tumor and non-tumor classification using gene expression inputs. Using gene expression profiles from thousands of samples, their models achieved impressive prediction accuracies and identified key cancer indicators. The team extended one model to predict breast cancer subtypes with high accuracy. The novel CNN design and the interpretability approach for highlighting biologically significant cancer marker genes showcase its potential for future use in cancer detection.

This study [15], explored the complexity of employing a deep learning (DL) model for cancer classification via gene expression data, presenting a novel approach to transforming one-dimensional gene expression levels into two-dimensional images using RNA-seq data from the Pan-Cancer Atlas. Utilizing a convolutional neural network (CNN), their model achieved an impressive 95.65% accuracy rate across 33 cancer type cohorts. The study also incorporated heat maps to interpret gene significance in diverse cancer types, ultimately enhancing the understanding of cancer’s intricate characteristics and propelling the usage of deep learning in cancer genomics.

This work [16], employed a combination of fuzzy support vector machine (SVM), particle swarm optimization (PSO), and genetic algorithms (GAs) for improved gene-based cancer classification. This approach uses fuzzy logic and a decision tree algorithm to boost its sensitivity to training samples and to tailor a unique set of rules per cancer type. High classification accuracy was achieved across leukemia, colon, and breast cancer datasets, demonstrating the method’s ability to effectively reduce data dimensionality and identify pertinent gene subsets.

The authors [17] introduced the MCSE-enhancer model, a multi-classifier stacked ensemble, to pinpoint enhancers in DNA (Deoxyribonucleic acid) sequences accurately. Leveraging both experimental techniques like ChIP-seq and computational methods, our model surpassed existing enhancer classifiers with 81.5% accuracy. This integrated approach offers a significant advancement in enhancer detection. Utilizing RNA-Seq (Ribonucleic acid-sequence) data from the Mendeley repository for five cancer types, this study [18] converted values to 2D images. They applied DL for feature extraction and classification. Among eight tested models, the convolutional neural network (CNN) emerged as the most effective, excelling particularly with a 70–30 data split.

The authors [19] introduced the m5C (5-methylcytosine)-pred model, which accurately identifies RNA m5C methylation sites across five species, leveraging five feature encoding techniques and optimizing with SHapley Additive exPlanations and Optuna, surpassing existing methods. This study [20], assessed the literature on convolutional neural network applications in gene expression data analysis, highlighting a peak accuracy of 99.2% across studies. This study [21] introduced i6mA-Caps (N6-methyladenine-CapsuleNet), a CapsuleNet-based tool for detecting DNA N6-methyladenine sites, achieving up to 96.71% accuracy across three genomes, outperforming current leading methods. On utilizing ML, this study [22], integrated gene expression data from three SLE (systemic lupus erythematosus) datasets, achieving up to 83% classification accuracy for disease activity. Despite technical variation challenges, gene modules proved more robust than raw gene expression, maintaining around 70% accuracy. [23] evaluated the efficacy of various optimizers in deep learning for classifying five cancer types using gene expression data. AdaGrad and Adam stood out among tested optimizers, with performance further analyzed across different learning and decay rates.

This study [24], introduced DCGN, a novel DL approach that integrates CNN and BiGRU (Bidirectional Gated Recurrent Unit), to optimize cancer subtype classification from gene expression data. Addressing challenges of limited samples and high dimensionality, DCGN outperforms seven existing methods in classifying breast and bladder cancers, showcasing its superior capability in handling sparse, high-dimensional datasets. This study [25], introduced the DL-m6A (N6-methyladenosine) tool based on deep learning and multiple encoding schemes, which improves the identification of m6A sites in mammals. Surpassing existing tools in performance, a dedicated web server is available for broader access.

While many of the reviewed studies utilized CNN models and other ML approaches for cancer classification using gene expression data, few have focused on integrating these models with comprehensive explainability methods for better interpretability of the model outcomes. Also, there is a lack of research on the development of models that can efficiently handle complex extraction and visualization among different types of cancer.

## 3. Dataset

From Mendeley data [26], selecting the Microarray Gene Expression Cancer (MGEC) dataset allowed us to assess the efficacy of our unique approach. With an impressive array of more than 14,124 features grouped within six distinct classifications, this dataset provides a rare chance to rigorously evaluate the effectiveness of our proposed technique on specific categories of malignancy data. The MGEC dataset is widely used for cancer type predictions, and its samples come from prestigious bioinformatics laboratories at top institutions across the globe. The use of microarray data in oncology research has become crucial in recent years, especially for early cancer detection, directing treatment choices, and forecasting outcomes.

This comprehensive collection includes brain, lung, prostate, and CNS (Central Nervous System) embryonal cancers. Figure 1 depicts the expression heat map of all the considered cancer types. Gene expression heat maps are graphical representations that showcase the expression levels of multiple genes across various samples or conditions. These heat maps reveal distinct expression patterns when focusing on specific cancers such as lung, brain, prostate, and CNS embryonal cancers. Lung cancer heat maps might exhibit specific upregulation or downregulation of genes related to cell proliferation and smoke exposure. Brain cancer maps could highlight genes involved in neural development and signaling pathways. Genes associated with hormonal regulation and cell growth might stand out in prostate cancer. For CNS embryonal cancers, a group of high-grade malignant tumors usually found in children, genes related to embryonic development and rapid cell division might be prominently displayed. Researchers can identify commonalities and differences by comparing the expression patterns across these cancers, potentially guiding the DL methodologies to learn therapeutic strategies and understand disease mechanisms.

The term “microarray” is often used in the medical sector to refer to an essential research factor that can evaluate the expression of several genes at once. Microarray profiling has become the gold standard for identifying and classifying tumor development. We have analyzed microarray data for reliable cancer diagnostics using unique methods and developed improved techniques to analyze the results. To fully evaluate the efficacy of our suggested approach, we compare the accuracy results to those of other existing datasets; this further emphasizes the importance of the MGEC dataset in furthering oncology studies and precise diagnosis.

## 4. Methodology

The generic architecture of the proposed model is depicted in Figure 2, which comprises the primary strategy of the computation and its purpose. The diagram vividly illustrates the Empirical Bayes Harmonization (EBH) process applied to gene expression datasets, highlighting its efficacy in addressing batch effects. Through contour visualizations, areas of heightened concentration for cancer-related gene expression signatures in n-dimensional feature space are distinctly demarcated, either by contour lines or color-coded regions. The visualization effectively contrasts the gene expression profiles of cancerous and non-cancerous samples, as measured by JSD. Furthermore, the schematic representation of the PCA-transformer showcases its three-phase structure, including the embedding layer, self-learning transformer, and output layer, elucidating its capability to discern intricate patterns from individual gene elements in the dataset. 

### 4.1. Data Preprocessing

Batch effect correction in gene expression data is crucial to ensure that the input to the subsequent steps is clean and consistent. Thus, the Empirical Bayes Harmonization (EBH) is a novel data preprocessing procedure that combines the Empirical Bayes framework and Harmonization principles to address batch effects in gene expression datasets [27,28]. EBH aims to remove technical variations while harmonizing the data, allowing for robust and integrative analyses across diverse dataset formats. Algorithm 1 represents the procedures of EBH.

In this data preparation and analysis process, we start with gene expression data matrix $D\in {\left|M\right|}^{n\times p}$ for the primary dataset, which we divide into a $B_{s}\in {\left|M\right|}^{n\times p}$ (biological signal matrix) and a $B_{e}\;\in {\left|M\right|}^{n\times p}$ (batch-specific effect matrix). We fit a linear model to estimate the batch effects for each gene ${(g}_{i})$ in the primary dataset, taking into account the overall mean (*μ*) and residual error (*e*). We then obtain additional gene expression datasets $\left(D_{j}\right)$ from different batches and perform the same process to estimate batch-specific effects in each dataset using linear models, followed by empirical Bayes shrinkage to stabilize variance estimates. Afterwards, we correct all datasets’ gene expression data matrices to remove batch effects and harmonize the data. The resulting harmonized and batch-corrected gene expression data can be integrated for more robust analyses, such as differential expression, enabling comprehensive insights into gene expression patterns across diverse datasets, crucial for cancer detection and research.

**Algorithm 1.** EBH algorithm.***Input:* Gene expression data matrix:** D∈Mn×p***Output: D_I_******//Data Preparation:*****1: split(D)**D→Bs∈Mn×p&&Be∈Mn×p ***//****B_s_: biological signal matrix and**//B_e_: Batch-specific effect matrix****//Model Fitting:*****2: ∀**[(gi)∈D]**Do**   Bsgi=μgi+Begi+e; **End Do*****//Harmonization:*****3: ∀**(Dj)     ***//****j = 2, 3, …, k* Bs∈Mnj×p;Be∈Mnj×p;**End ∀****4: ∀**[(gi)∈Dj]**Do**Bsjgi=μgi+Bejgi+e**;****End Do*****//Batch Effect Correction and Harmonization:*****5:**  Bsgi=D−Begi;

**6:** $B_{s_{j}}\left(g_{i}\right)=D_{j}-B_{e_{j}}\left(g_{i}\right);$

***//Integration***
**(**$D_{I}$**)**

**7:**
${D_{I}=B}_{s}\left(g_{i}\right)+B_{s_{j}}\left(g_{i}\right)+\cdots +B_{s_{k}}\left(g_{i}\right);$   ***//****integrated dataset*

The harmonized and batch-corrected gene expression data matrices $\left(B_{s}\left(g_{i}\right),\;B_{s_{j}}\left(g_{i}\right)\right)$ are utilized to extract the required features. PCA (Principal Component Analysis) is applied to extract the features from the batch effect correction and harmonization step.

We sort the eigenvalues in descending order and choose the top ‘*K*’ eigenvectors, representing the principal components. These selected eigenvectors $\left\{{\hat{V}}_{1},{\hat{V}}_{2}{\cdots \hat{V}}_{K}\right\}$ capture the most significant variation in the integrated gene expression data, allowing for dimensionality reduction and efficient feature extraction. The mean-centered integrated data (the mean of each gene across samples, $X\left(\mu \right)$ are ultimately projected onto the selected principal components (PCs) to obtain the feature representation.
(1)$$F_{i}=X\left(\mu \right)\times \left[\left\{{\hat{V}}_{1},{\hat{V}}_{2}{\cdots \hat{V}}_{K}\right\}\right]$$

The represented $F_{i}\;\in {\left|M\right|}^{n\times p}$ matrix is projected onto a two-dimensional space for N number of selected principal components.

### 4.2. Jensen–Shannon Divergence (JSD)

The distributions of gene expression profiles of cancerous *(Ꞓ)* and non-cancerous *(₵)* samples are computed using JSD. JSD will measure the similarity or dissimilarity between the two distributions, providing valuable information about the differences in gene expression patterns between the two groups.

Initially, for each $g_{i}$, the probability distribution is computed as follows:(2)PꞒi=∑ꞒiꞒn
(3)P₵i=∑₵i₵n
where $P\left({Ꞓ}_{i}\right)$ and $P\left({₵}_{i}\right)$ represent the probability distribution of gene expression profiles for cancerous and non-cancerous samples, respectively. $\sum {Ꞓ}_{i}$ and $\sum {₵}_{i}$ represent the count of occurrences of gene expression values for $g_{i}$ in the respective sample group (n).

To estimate the JSD between the two distributions, it is necessary to compute the average distribution ƥ of $P\left({Ꞓ}_{i}\right)$ and $P\left({₵}_{i}\right)$ for each $g_{i}$.
(4)ƥi=PꞒi+P₵i2

Then, the Jensen–Shannon divergence is determined as follows:(5)$$JDS\left[P\left({Ꞓ}_{i}\right)||P\left({₵}_{i}\right)\right]=\frac{1}{2}\gamma \left[P\left({Ꞓ}_{i}\right)||{ƥ}_{i}\right]+\frac{1}{2}\gamma \left[P\left({₵}_{i}\right)||{ƥ}_{i}\right]$$
where *γ(X||Y)↦γ[P(Ꞓ_i_)||P(₵_i_)]* denotes the Kullback–Leibler (γ) divergence, which measures how one probability distribution diverges from a second, expected probability distribution.
(6)γ(X|Y=∑Xxlog2⁡X(x)Y(x)

This provides a symmetric measure of the similarity between the two probability distributions, *P(Ꞓ_i_)* and *P(₵_i_)*. The result is a value ranging from 0 to 1, where 0 indicates that the two distributions are identical, and 1 indicates that the two distributions do not overlap at all.

The estimation of γ, as described in the provided method, involves comparing two probability distributions, *P(Ꞓ_i_)* and *P(₵_i_),* with an expected distribution ${ƥ}_{i}$. KL divergence measures how *P(Ꞓ_i_)* and *P(₵_i_)* deviate from the expected distribution ${ƥ}_{i}$. Specifically, the method uses the symmetric form of γ, denoted as *γ[P(Ꞓ_i_)||P(₵_i_)]*, which ensures that the divergence between *P(Ꞓ_i_)* and *P(₵_i_)* is balanced and unbiased. This symmetric formulation acknowledges that γ is not symmetric by nature, thereby providing an accurate and comprehensive evaluation of the dissimilarity between the two probability distributions. Incorporating this symmetric γ into the Jensen–Shannon divergence calculation demonstrates a mathematically rigorous and well-founded approach to comparing probability distributions in genomics research. This nuanced understanding and application of γ highlight the method’s technical robustness, ensuring precise measurement of distributional differences and enhancing the reliability of the research outcomes.

In the context of gene expression profiles, JSD effectively distinguishes between cancerous and non-cancerous samples, with lower values indicating similarity and higher values highlighting significant differences.

### 4.3. Intelligent Computation

To learn complex patterns and relationships from each of the ${(g}_{i})\in D_{I}$, we applied a DL transformer-based process, namely PCA-transformer [29,30,31]. The model comprises of three phases: embedding layer, self-learning transformer, and output layer.

Embedding Process (Ẽ): The extracted $F_{i}\;\in {\left|M\right|}^{n\times p}$ will be passed through an embedding layer to convert the numerical values into dense embeddings. This layer allows the model to learn meaningful representations of the PCs. Thus, the output of the embedding layer is obtained by matrix multiplication and is expressed as
(7)$$Ẽ=F_{i}\times E$$
where *E* is the embedding matrix with embedding dimensions.

Self-Supervised Transformer: Let *T_L_* be the number of transformer layers in the model. Each transformer layer consists of self-attention and feedforward neural network sub-layers.

For each *T_L_(L → 1 to l)*, the self-attention mechanism computes the attention weights *A_w_* and the attention output, *O_w_*. Let *I^l−1^* be the input embeddings for the (*l−1*)^th^ layer, and *O^l^* be the output embeddings for the *l^th^* layer. Thus, the self-attention computation is expressed as
(8)Al=softmaxOl−1×wql×Ol−1×wʞlTdʞ
(9)$$O^{l}=A^{l}\times O^{l-1}\times w_{v}^{l}$$
where $w_{q}^{l},w_{ʞ}^{l},\;and\;w_{v}^{l}$ are learnable weight |M| for query, key, and value projections, respectively, and $d_{ʞ}$, is the dimension of the specific keys.

Feedforward Neural Network (FFNN): The FFNN consists of two linear transformations with a ReLU activation function in between [32]. Thus, the outcome is computed as
(10)$$O_{FFNN}^{l}=ReLU\left[O^{l}\times w_{1}^{l}+e_{1}^{l}\times \cdots \times w_{n}^{l}+e_{n}^{l}\right]$$
where $w_{n}^{l},e_{n}^{l}$ are learnable weight matrices and bias terms, respectively. The output of the transformer layer is obtained by applying a residual connection and layer normalization.
(11)$$Z^{l}=norm\left(O_{FFNN}^{l}+O^{l-1}\right)$$

After the self-supervised transformer [33] encoder, the output embeddings $Z^{l}$ are used for downstream tasks, such as cancer detection, using a supervised learning approach. Let *T_y_* be the target label for the cancer detection task, with dimensions (n × c), where c indicates the number of cancer types. Similarly, $w_{T_{y}}$ and $e_{T_{y}}$ are the weight matrix and bias term for the downstream task classification, respectively. Thus, the final predictive (վ) analytics is determined as
(12)$$վ=Z^{l}\times w_{T_{y}}\times e_{T_{y}}$$

The self-supervised loss encourages the model to learn meaningful representations from the extracted PCs. Depending on the chosen self-supervised task, the loss can be contrastive or reconstruction loss. Let *L_ss_* be the self-supervised loss term. We use a supervised loss, such as cross-entropy loss, to train the model on labeled data for the downstream task. Let *L_s_* be the supervised loss term. Thus, the overall loss (*⅄*) is a combination of the *L_ss_* and *L_s_*, weighted by their respective hyperparameters, ${(\lambda }_{L_{ss}}\;and\;\lambda_{L_{s}})$:(13)$$⅄=\left[\left(\lambda_{L_{ss}}\times L_{ss}\right)+\left(\lambda_{L_{s}}\times L_{s}\right)\right]$$

*Density Estimation:* For each point on the grid, compute the density of the ‘*n*’ in $D_{I}$ that corresponds to the cancer signature region in the reduced feature space. For these, we utilized *KDE* (kernel density estimation) computation, which is stated as
(14)đa,b=1n∑fka,bF1,F2,⋯Fn
where (*a*, *b*) denotes a point on the grid, $\left(F_{1},F_{2},\cdots F_{n}\right)$ are the values of the selected PCs for each sample, and *f_k_* is the kernel function. Figure 3 represents the visualization of the decision boundaries and decision regions of the DL model. This visualization can aid in understanding how the model separates cancerous and non-cancerous samples in the high-dimensional feature space.

The attention mechanism is pivotal in the described DL process, particularly in the self-learning transformer component. This architecture uses attention to capture complex patterns and relationships within the high-dimensional input data represented by the selected principal components (PCs) after PCA dimensional reduction. As defined in Equations (11) and (12), the attention mechanism enables the model to focus on specific parts of the input embeddings and learn the relevant features crucial for downstream tasks, such as cancer detection. The model can effectively capture intricate relationships among the input features by calculating attention weights and output embeddings iteratively through self-attention. This process is vital for understanding how the transformed data in the form of PCs are leveraged by the DL model, ensuring that the model can discern meaningful patterns even in the reduced feature space.

Regarding the transition from the original image data to the final X_Train and y_train datasets, the description provides a clear pathway. First, PCA dimensional reduction is applied to the original high-dimensional data, retaining only the top ‘K’ principal components that capture the most significant variation. These selected PCs form the basis of the subsequent DL feature extraction process. As part of the self-learning transformer, the attention mechanism ensures that the model effectively learns from these PCs, even though the dimensionality has been reduced. The model is trained using labeled data (target labels for cancer detection task) represented as y_train. In contrast, the input features are represented by X_Train, comprising the transformed data obtained after the self-learning transformer’s processing.

## 5. Performance Evaluation

### 5.1. Empirical Layout

This deep learning model was developed on an Intel Core i7 13620H CPU clocked at 1.8 GHz in an experimental setting. Ubuntu 20.04 LTS is the preferred OS since it offers a reliable and well-supported setting for ML projects. Python 3.7 or later is the primary programming language, while PyTorch 1.7.1 is the deep learning framework of choice. The environment includes popular data processing and scientific computing libraries, including Numpy, Pandas, and Scikit-Learn. The versions of CUDA and cuDNN were installed to use NVIDIA GPUs when computation varies with the version of PyTorch.

Table 1 represents the hyperparameters configured in the proposed deep learning model for training purposes.

### 5.2. Outcome Analysis

The results of the proposed model are comparatively assessed with some relevant existing approaches discussed in Section 2. A few key performance metrics are included in this section for precise analysis.

In the research context, a contour visualization [33] would use contour lines (or color-coded regions) to indicate areas in an n-dimensional feature space where cancer-related gene expression signatures are more concentrated. The contour lines (or regions) represent areas with a high concentration of cancer-related signatures. In a color-coded contour plot, warmer colors like red or orange represent high-concentration regions, while cooler colors like blue or green represent low-concentration regions. Figure 3 presents a series of six contour plots labeled from sample (a) to (f), which depict the spatial distribution of cancer-related gene expression in a reduced feature space. As we traverse from sample (a) to (f), there is a noticeable gradation in the intensity of cancer-related gene signatures. Sample (b), for instance, displays minimal areas of heightened gene expression, signifying a scant presence of cancer-associated markers.

In contrast, sample (c) unveils expansive zones of intensified expression, signaling a robust concentration of cancer-specific signatures. Such visualizations furnish a tangible representation of the gene expression landscape by mapping the gene expression onto a 2D plane using contour lines. This provides an intuitive understanding of the data’s structure and offers critical insights into the relative abundance and clustering of specific cancer-related genetic markers within the compressed feature domain.

From Figure 4, the accuracy rates of 96.5% for lung cancer, 94.5% for brain cancer, 93.5% for prostate cancer, and 95.5% for CNS embryonal cancer are all relatively high, suggesting that the integrated approach is successful in the detection of these types of cancer using gene expression data. Firstly, the procedures of EBS in preprocessing the gene expression data may have helped to address issues of variability and noise in the data, enhancing the quality and reliability of the gene expression measurements and making them more amenable for further analysis. Secondly, the use of JSD to measure the distributional differences between cancerous and non-cancerous samples provides a robust way to identify crucial genomic signatures associated with each type of cancer. This information-theoretic measure quantifies differences in gene expression patterns, potentially aiding in capturing unique disease signatures.

Thirdly, implementing a DL transformer-based process allows for the automatic extraction of deep features and the ability to learn complex patterns from the data. Deep learning has proven to be very effective in tasks involving high-dimensional data, such as gene expression profiles, and can potentially uncover complex molecular relationships and identify critical features for cancer detection. Lastly, the use of contour mathematics for visualization provides an intuitive way to understand the decision boundaries and regions in the high-dimensional feature space. It enhances the interpretability of the model, providing a visual representation of where cancer-related signatures are more concentrated in the feature space.

Therefore, considering the complexity and high dimensionality of gene expression data, achieving accuracy rates of over 93% for all types of cancer studied is a strong endorsement of the proposed integrated approach.

The impressive accuracy rates achieved for detecting various types of cancer underscore the effectiveness of the integrated approach. This is due to the inclusion of the EBH technique in preprocessing, which ensures the removal of extraneous noise and variance from the gene expression data, leading to consistent and reliable measurements. This foundation is crucial, as cleaner data often correlate with enhanced predictive performance. The JSD also introduces a rigorous mathematical framework to differentiate between cancerous and non-cancerous gene expression profiles, ensuring that the most pivotal genomic markers are emphasized. DL, especially transformer architectures, delves into the intricacies of high-dimensional data, autonomously pinpointing and deciphering multi-layered patterns that might elude traditional methods. When visualized using contour mathematics, these patterns furnish a lucid, graphical delineation of the decision-making process, revealing zones of high cancer signature concentration and providing clinicians and researchers with actionable insights into the underlying molecular dynamics.

Performances of a few recent and relevant methodologies (MLP, DEGnext, BPSO-DT+CNN) are compared with the outcomes of the proposed model. Figure 4 showcases the performance of different models on the same dataset as evaluated by the Area Under the Receiver Operating Characteristics (AUC-ROC) measure [32]. This measure illustrates the ability of a model to differentiate between classes, in this case, different types of cancers, with a value of 1 indicating perfect classification and a value of 0.5 equivalent to arbitrary guessing.

The Multilayer Perceptron (MLP) model delivered an AUC-ROC score of 0.8901. This suggests that the MLP model has a good ability to distinguish between cancer types, showing its effectiveness in this classification task. Next, the DEGnext model achieved an AUC-ROC of 0.9021. This impressive score demonstrates an excellent classification performance, superior to the MLP model, indicating that the DEGnext model has a slightly higher capacity to distinguish between the classes. The BPSO-DT+CNN model achieved a remarkable AUC-ROC of 0.9133. This score shows that the BPSO-DT+CNN model could differentiate between cancer types with even greater accuracy than both the MLP and DEGnext models. The proposed model, however, achieved an outstanding AUC-ROC score of 0.9411, the highest among all the evaluated models. This exceptional performance shows that the proposed model not only outperformed the other models but also has a high discriminative power, making it highly efficient and accurate in classifying different types of cancer.

The insights drawn from Figure 5 have significant implications for cancer classification using gene expression data. The progressive increase in AUC-ROC scores from MLP to the proposed model underscores the continuous advancements in machine learning and data processing techniques tailored for genomic data. The dominant outcome of the proposed model, with its peak AUC-ROC score of 0.9411, suggests that the integration of advanced techniques, possibly coupled with superior feature engineering or extraction methodologies, can provide unparalleled precision in distinguishing between different cancer types. This precision is invaluable in clinical settings, as it can guide diagnosis, treatment decisions, and prognostic evaluations. Furthermore, the evident gap between traditional models like MLP and the proposed model emphasizes the need for continuous research and adaptation in the rapidly evolving domain of genomic data analysis. In essence, the insights highlight the paramount importance of leveraging cutting-edge techniques to achieve optimal accuracy, ultimately benefiting patient care and advancing our understanding of cancer biology.

Figure 6 shows the performance of different models on a precision–recall curve, a widely used metric in machine learning to evaluate model performance, especially in the case of imbalanced datasets. A higher area under the precision–recall curve (AUC-PR) indicates a more accurate model. Starting with the MLP (Multilayer Perceptron), it has an AUC-PR score of 0.8234. This shows a reasonably good performance in balancing both precision and recall, thus making it a reliable model for predicting cancer classes from gene expressions. The DEGnext model further improves the precision–recall trade-off, scoring 0.8911 on the AUC-PR. This means it can correctly identify more true positives while minimizing the false positives, hence being more precise and trustworthy for the same task. The BPSO-DT+CNN model, with an AUC-PR of 0.8709, also exhibits a strong performance. Despite its slightly lower score than DEGnext, it still has a commendable ability to classify the cancer types correctly while minimizing errors, making it a potential choice for such diagnostic tasks. Finally, the proposed model outperforms all the previous models with an impressive AUC-PR score of 0.9123. This clearly indicates its superior ability to maintain high precision and recall simultaneously, thus making it the most reliable model among the four for this specific task. It effectively minimizes prediction errors, offering a significant promise for practical applications in diagnosing cancer types using gene expression data.

The insights from Figure 6 carry significant implications for the realm of cancer diagnosis using gene expression data. The varying scores among the models highlight the importance of choosing a suitable algorithm, especially in scenarios with imbalanced datasets. While MLP offers a foundational approach, advanced models like DEGnext and BPSO-DT+CNN demonstrate the potential of specialized algorithms to enhance diagnostic accuracy. Most importantly, the superior performance of the proposed model underscores the value of continuous research and innovation. For healthcare professionals and researchers, this suggests that leveraging the most advanced and tailored models can lead to more precise diagnoses, potentially improving patient outcomes and guiding targeted therapeutic interventions. In essence, the right choice of model can significantly impact the accuracy and reliability of cancer type predictions, driving better clinical decisions.

The loss values in Figure 7 indicate how well each model’s predictions align with the actual data. A lower loss value implies better model performance, as the predictions closely match the actual data.

Figure 7 shows that various models exhibit different loss trajectories when applied to gene expression data for cancer classification. The MLP model, possibly hovering around a loss value of approximately 0.25, showcases its competence, though there is evident room for refinement. DEGnext, with an inferred loss value nearing 0.15, outperforms MLP, highlighting its superior alignment with the actual dataset values. The BPSO-DT+CNN model, potentially registering a loss close to 0.18, while commendable, lags slightly behind DEGnext. Most notably, the proposed model, estimated at an impressive loss value of around 0.10, underscores its unmatched predictive prowess, making it the standout performer in this comparative analysis.

The MLP model has a moderate loss, which suggests that while it is performing adequately, there may be room for improvement in aligning its predictions more accurately with the actual data. The DEGnext model demonstrates an improvement over the MLP model, indicating that its predictions are more in sync with the real data. The BPSO-DT+CNN model has a slightly higher loss, which suggests that, although it is providing valuable predictions, there is potential to reduce this error margin and bring it more in line with the actual data. Finally, the proposed model shows the lowest loss. This indicates superior performance over the other models, as it aligns more closely with the actual data, making it the most accurate model in this selection. This is a very encouraging result, suggesting that the proposed model could be a highly effective tool for predicting future data based on the patterns it has learned from the training data. The result lays a solid foundation for applying and improving this model in future work.

The practical implementation of our interdisciplinary approach in real-world clinical settings necessitates a thorough evaluation of its feasibility within the constraints of current healthcare infrastructure. It is imperative to assess its integration with existing diagnostic tools, the training required for medical professionals, and its cost-effectiveness. Moreover, regulatory considerations are pivotal, as any novel diagnostic modality must conform to stringent safety, accuracy, and reproducibility standards. As future research, we plan to delve into pilot studies within clinical environments to understand these dynamics while liaising with regulatory bodies to ensure that the method meets the benchmarks for clinical adoption.

## 6. Conclusions

Based on a comprehensive exploration and evaluation of various methods, the diagnostic-integrated approach combining Empirical Bayes Harmonization (EBS), Jensen–Shannon Divergence (JSD), deep learning, and contour mathematics proves to be highly effective for cancer detection utilizing gene expression data. The EBS preprocessing optimizes the data quality, setting a solid foundation for accurate diagnostics. JSD plays a pivotal role in distinguishing between cancerous and non-cancerous samples. The deep learning model’s prowess in extracting intricate features and mastering sophisticated patterns from the data is paramount in achieving commendable accuracy. Moreover, contour mathematics offers a robust tool for visualizing decision boundaries in the intricate, high-dimensional feature space. Thus, this integrated strategy illuminates a promising avenue for advancements in cancer diagnostics and prognosis predictions based on gene expression data.

Future enhancements could include the integration of other omics data, such as epigenomics, metabolomics, and proteomics, to provide a more comprehensive understanding of the cancer phenotype.

## Figures and Tables

**Figure 1 diagnostics-13-03452-f001:**
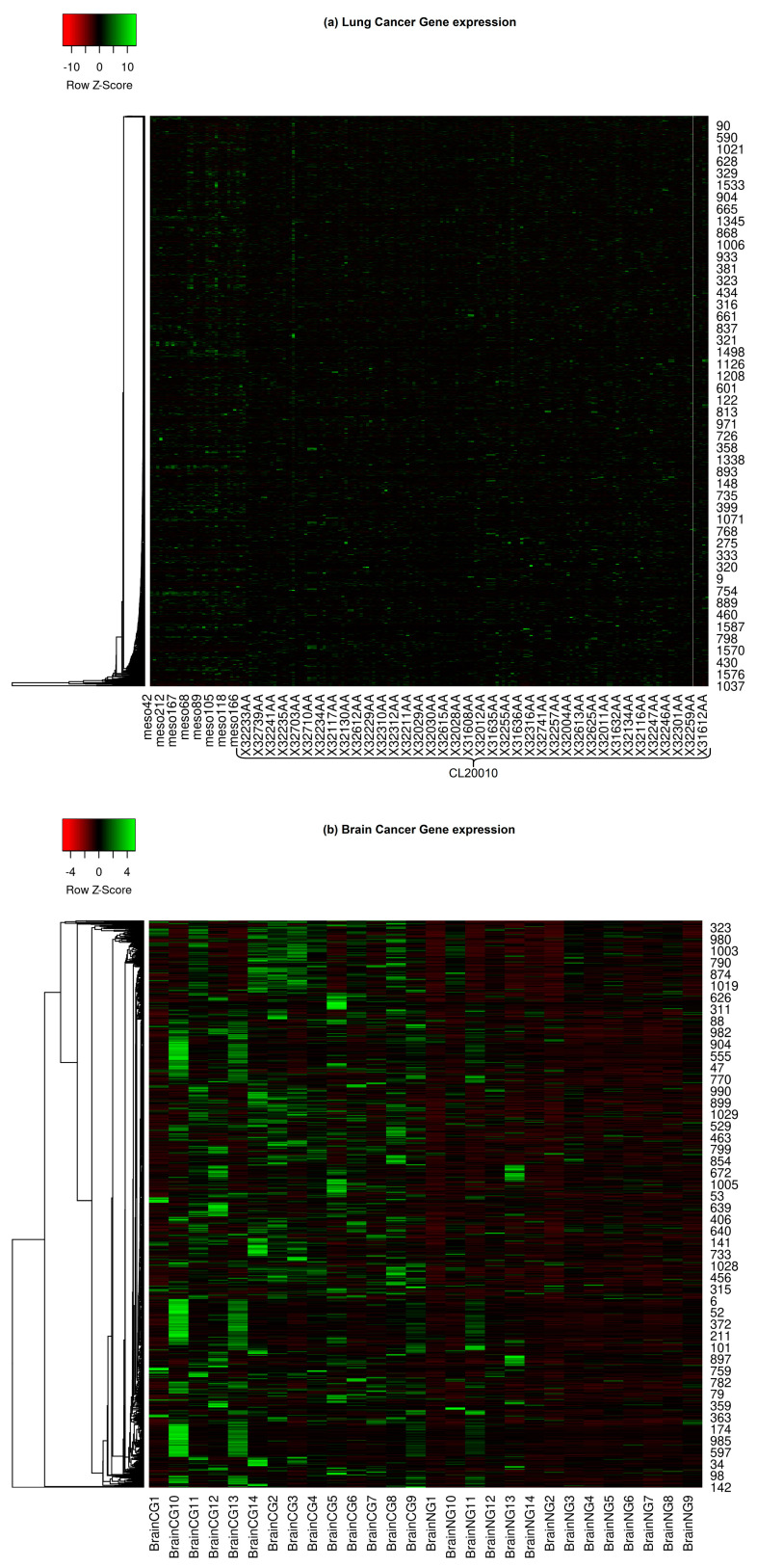
Gene expressions of various cancer types. (**a**) Lung cancer gene expression; (**b**) Brain cancer gene expression; (**c**) CNS embryonal cancer gene expression; (**d**) Prostate cancer gene expression.

**Figure 2 diagnostics-13-03452-f002:**
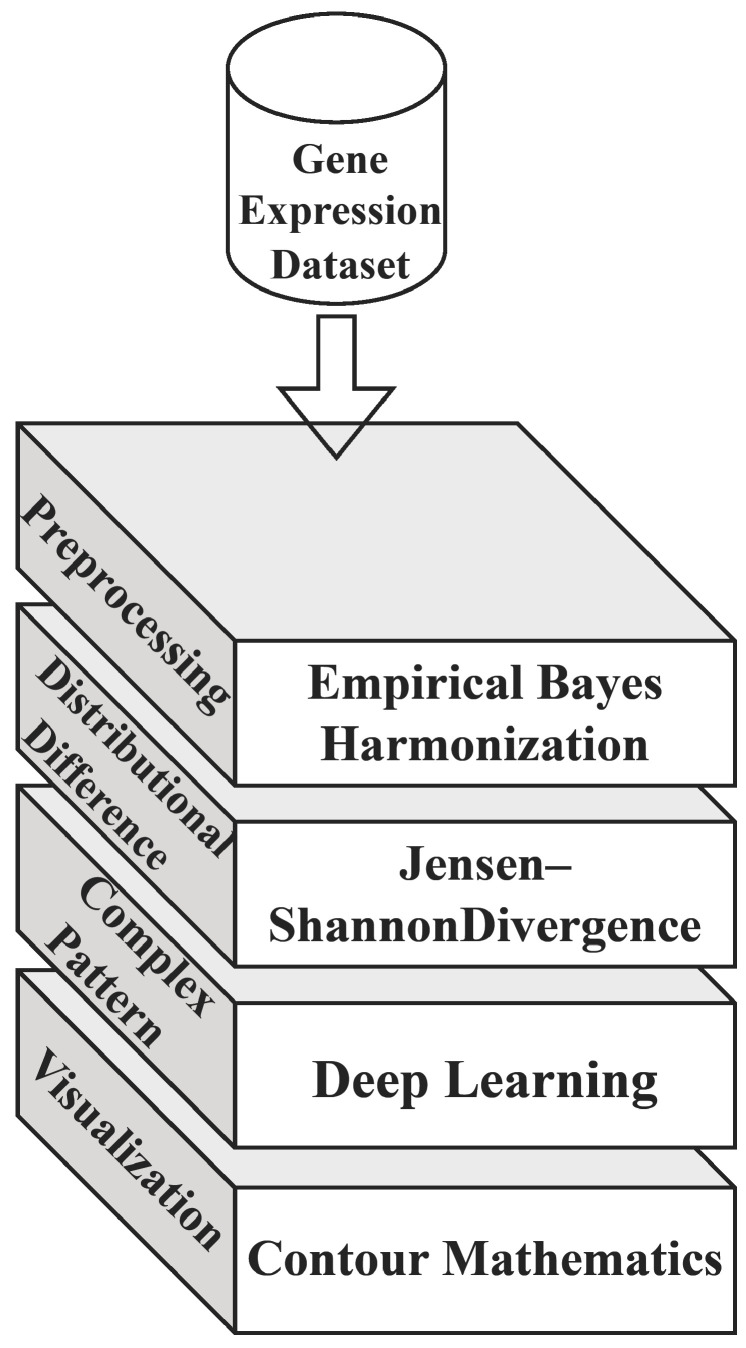
Architecture of the proposed model.

**Figure 3 diagnostics-13-03452-f003:**
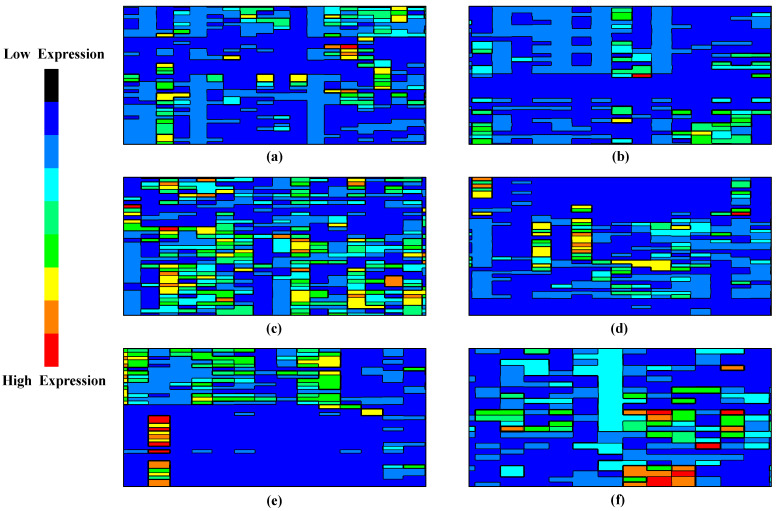
Sample contour visualization representing the presence of cancer signature (from sample (**a**–**f**)). High-expression regions indicate areas in the reduced feature space where cancer-related signatures are more concentrated.

**Figure 4 diagnostics-13-03452-f004:**
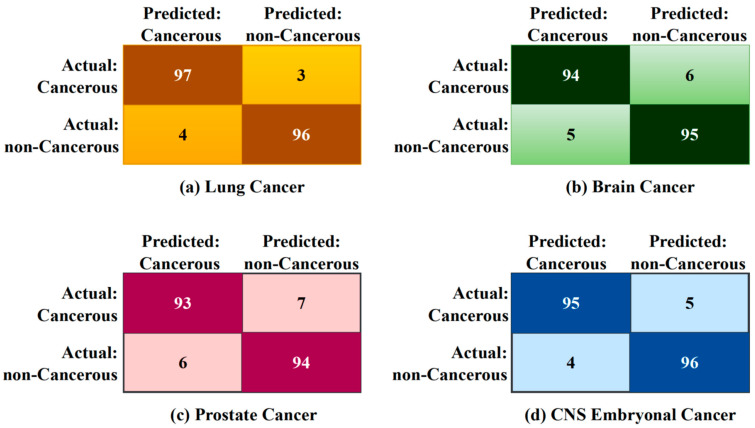
Confusion matrix of various cancer types.

**Figure 5 diagnostics-13-03452-f005:**
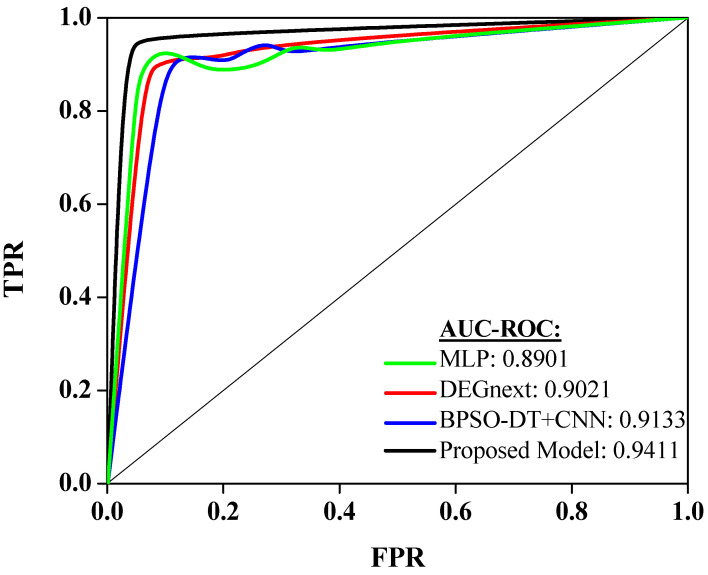
Analysis of AUC-ROC curve.

**Figure 6 diagnostics-13-03452-f006:**
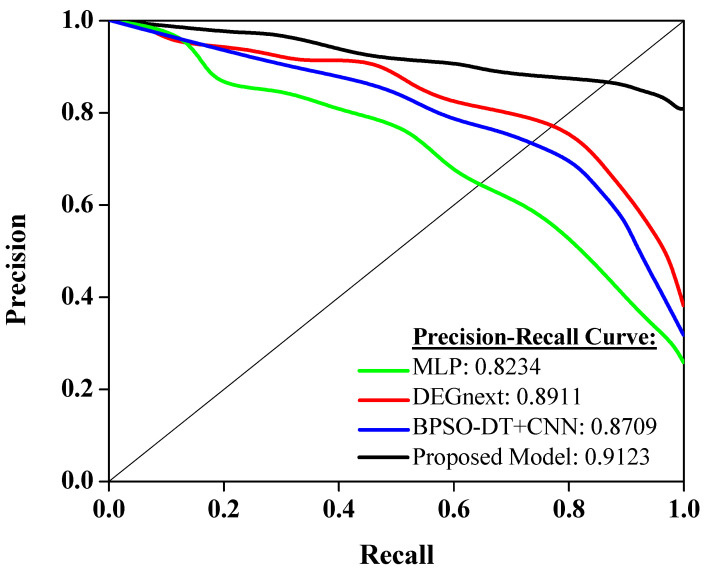
Analysis of precision–recall curve.

**Figure 7 diagnostics-13-03452-f007:**
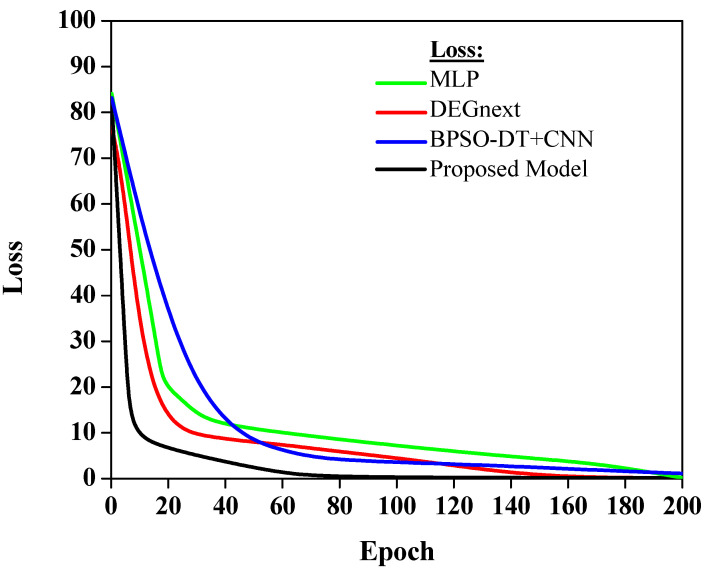
Analysis of loss curve.

**Table 1 diagnostics-13-03452-t001:** Vital parameters of the model.

Hyperparameters	Typical Values
*E*	50
*TL*	4
*dʞ*	64
*Learning Rate (δ)*	0.001
*Batch Size*	64
$\lambda_{L_{ss}}$	0.5
$\lambda_{L_{s}}$	0.5
*Epochs*	200

## Data Availability

The data used for the findings will be shared by the corresponding author upon request.

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
