# Peer review of "Cancer Diagnosis through Contour Visualization of Gene Expression Leveraging Deep Learning Techniques"

_diagnostics, 2023, doi:10.3390/diagnostics13223452_

Round 1

Reviewer 1 Report

Comments and Suggestions for Authors The authors presented a rather interesting paper on an integrated approach to gene expression-based cancer detection based on empirical Bayesian harmonization, Jensen-Shannon divergence, deep learning and contour mathematics. The authors demonstrated the high efficiency of this analytical approach to analysis compared to other methods. The authors' findings from four data sets across different cancer types highlight the promise of using this tool in cancer research. However, there is a note to the authors:   - I would recommend the authors to add a more complete description to Figures 1-2 and 4-7. - In addition, typos in lines 44, 50, 63, 66, 70, 77, 174, 203 and 345 need to be corrected. Comments on the Quality of English Language

The quality of the English language requires stylistic corrections

Reviewer 2 Report

Comments and Suggestions for Authors

The manuscript introduces an innovative approach for cancer detection by combining Empirical Bayes Harmonization (EBS), Jensen-Shannon Divergence (JSD), deep learning, and contour mathematics. This interdisciplinary approach is a strong point and can potentially contribute significantly to the field of cancer diagnostics. I would like the authors to address following concerns, 

- The manuscript mentions the potential applications of your method in clinical diagnostics and personalized treatment strategies. However, it does not provide insights into the practicality of implementing this approach in real clinical settings or the regulatory considerations associated with such applications.

- The manuscript lacks references to related studies or prior work in the field. Providing a background on the existing research in cancer detection and the shortcomings of current methods would help position your work in the context of the broader field. I suggest the authors to add following recent articles from literature and find more by themselves, 

1. Improving Enhancer Identification with a Multi-Classifier Stacked Ensemble Model

2. Analyzing RNA-seq gene expression data using deep learning approaches for cancer classification

3. XGBoost Framework with Feature Selection for the Prediction of RNA N5-methylcytosine sites

4. A review on convolutional neural network based deep learning methods in gene expression data for disease diagnosis

5. i6mA-Caps: A CapsuleNet-based framework for identifying DNA N6-methyladenine sites

6. Machine learning approaches to predict lupus disease activity from gene expression data

7. Deep learning techniques for cancer classification using microarray gene expression data

8. DL-m6A: Identification of N6-methyladenosine Sites in Mammals using deep learning based on different encoding schemes

9. Deep learning approach for cancer subtype classification using high-dimensional gene expression data

- There is a lack of information about the availability of the code, models, or datasets used in the research. This can hinder the reproducibility and transparency of the study, a critical aspect of scientific research.

- I have a query regarding title. Whether the title accurately reflects the content of the manuscript, given the limited technical details provided in the current version? I think it can be improved. 

- Ensure that you provide proper citations for the methods and techniques mentioned in this section, such as EBH, JSD, and PCA-transformer, to give credit to the original sources and allow readers to explore them further.

- Provide a brief context or rationale for the choice of these specific methods. Why were EBH, JSD, and the PCA-transformer chosen for this study? What advantages do they offer over alternative approaches?

Comments on the Quality of English Language

I think its fine

Reviewer 3 Report

Comments and Suggestions for Authors

The abstract is too long, repeting the same idea several times.

The related work section is a compilation of related papers in different paragraphs while it should be an explanation of how your work is integrated in previous works.

Fig 1 requires a better explanation of the heatmap or legend (green/red colors) for people not expert on gene expression to understand the problem from a classification/regression point of view.

There are two Figure 1.

Figures are not good looking. In a scientific paper you do not need so fancy colors or graphs if not necessary, but to be more effective in summarizing the proposed algorithm and when it is really necessary. And you do not need so much space. For example, Fig. 2 is irrelevant when you can explain it in the text.

Table 1 is very hard to read. Formatting is awful.

Review some bugs such as "is applied toobtain to extract".

You do not need to explain very basic techniques such as PCA. Focus in the usefulness and from an algorithmic point of view, how the input is transformed in a vew input for the DL deature extraction.

How do you estimate the Kullback-Leibler divergence? By the way, it is not symmetric (it is not a distance).

This is a typical example of long paragraph plenty of repetitions and obvious explanations: "JSD is an effective method for comparing the similarity or dissimilarity of two distributions, especially in the context of gene expression profiles. In this scenario, a lower JSD value between cancerous and non-cancerous samples would suggest a similar gene expression pattern, while a higher JSD value would indicate significant differences in the gene expression profiles of the two groups".

The same for the DL part of the paper. Instead of explaining the attention mechanism, explain better how you come from the original image to the X_Train and y_train datasets (how PCA dimmensional reduction affects, how you can get a good result with how many images? dimensions and architecture? and other related DL issues). I do not see any of this analysis in the results section.

Round 2

Reviewer 2 Report

Comments and Suggestions for Authors

I think authors have addressed all my queries.

Author Response

Dear Reviewer

Thanks for giving the acceptance.

Thanks

Reviewer 3 Report

Comments and Suggestions for Authors

I think authors misunderstood some of my comments. I was not asking for explanations about why PCA or attention mechanism are used or useful. This was clear; I was recommending to shorten this basic knowledge, but authors prefer to keep all these well known theory including the equations.

And I do not agree with keeping the Figures 2 and 3 as they were. Frankly, I am not used to see Fig.3 kind of figures in a machine learning-based paper (nowadays you can find references supporting almost anything; I recognize that figures are very important to ease the understanding of a "hard" paper, specially block diagrams or examples, but this is not the case, specially when you refer to other previous works doing similar things). 

Round 3

Reviewer 3 Report

Comments and Suggestions for Authors

It is not a problem of colors in the figures, but I appreciate your efforts